# Systems Biology: New Insight into Antibiotic Resistance

**DOI:** 10.3390/microorganisms10122362

**Published:** 2022-11-29

**Authors:** Piubeli Francine

**Affiliations:** Department of Microbiology and Parasitology, Faculty of Pharmacy, University of Seville, 41012 Seville, Spain; piubeli@us.es

**Keywords:** antibiotic resistance, omics approches, system biology, mathematical models, genome-scale metabolic models

## Abstract

Over the past few decades, antimicrobial resistance (AMR) has emerged as an important threat to public health, resulting from the global propagation of multidrug-resistant strains of various bacterial species. Knowledge of the intrinsic factors leading to this resistance is necessary to overcome these new strains. This has contributed to the increased use of omics technologies and their extrapolation to the system level. Understanding the mechanisms involved in antimicrobial resistance acquired by microorganisms at the system level is essential to obtain answers and explore options to combat this resistance. Therefore, the use of robust whole-genome sequencing approaches and other omics techniques such as transcriptomics, proteomics, and metabolomics provide fundamental insights into the physiology of antimicrobial resistance. To improve the efficiency of data obtained through omics approaches, and thus gain a predictive understanding of bacterial responses to antibiotics, the integration of mathematical models with genome-scale metabolic models (GEMs) is essential. In this context, here we outline recent efforts that have demonstrated that the use of omics technology and systems biology, as quantitative and robust hypothesis-generating frameworks, can improve the understanding of antibiotic resistance, and it is hoped that this emerging field can provide support for these new efforts.

## 1. Introduction

Antibiotics have been an indispensable part of healthcare for years, enabling the treatment of diseases caused by previously fatal bacterial infections [1]. Although antibiotics remain essential in health care, their effectiveness has been compromised by the rapid rise in the occurrence of resistance to them. Considering the important role of antibiotics in society and the mechanisms and factors influencing antibiotic efficacy, there is an urgent need for improved understanding.

The main targets and mechanisms of action of most antibiotics have been identified and are well-studied [2]; however, it is increasingly evident that antibiotic efficacy is a highly complex process occurring at the system level, where a network of events involving both the mechanism of action of antibiotics and the physiology of stress generated by the presence of the antimicrobial must be considered [3]. In this sense, systems biology is an approach that is helping to improve the understanding of the processes involved in the acquisition of resistance. Further, systems biology can be defined as the approach that integrates different levels of omics technologies (i.e., from DNA, through RNA, proteins, and metabolites to phenotypes) with genome-scale metabolic models (GEMs). The amalgamation of these approaches has enhanced our understanding of a complex process in exquisite molecular detail [4,5,6] The use of this novel approach allows us to obtain information on biological questions, build testable hypotheses, and foster the engineering development of new technologies and novel strategies to address these emerging challenges related to antibiotic resistance (AR) [7].

By approaching systems biology as a tool in the investigation of antibiotic-related questions, a holistic view is obtained that allows us to go beyond the traditional restrictive search focused on essential targets, thus being able to obtain a global view of antibiotic-related resistance mechanisms, and the consequences of their acquisition for bacterial cellular function.

The richness of using omics approaches is that they can provide answers on their own and also when integrated with other omics or GEMs. Genomics, for example, enables the discovery of novel resistance-related genes and the study of the phylogeny of isolates, in addition to epidemiological traits, which are critical to help understand AR [8]. On the other hand, the transcriptomic approach allows us to observe global changes that occur in gene expression because of environmental variations, and are therefore highly appropriate for providing a comprehensive image of microbial responses to antibiotic therapy. In turn, proteomics, to a great extent, has been applied to further the comprehension of the steady-state proteome of antibiotic-resistant clinical strains. Finally, metabolomics allows us to obtain a global view of all the molecules involved in metabolic processes at each state, which in turn is intrinsically related to the phenotype of the organism. 

In this framework, the usefulness of data obtained from different omics can be maximized by integrating them into GEMs, either together or separately. Such integration, based on the superimposition of omics data on predefined metabolic models to understand molecular organization and function [9], facilitates the design of new experiments and promotes more accurate predictions by providing robustness and precision to the model. The great interest of this approach, in particular the use of GEMs, lies in the ability to perform simulations that incorporate genome-wide data and can even add custom constraints; this includes adjusting the flux through particular reactions (e.g., simulating elimination) or the presence of specific nutrient requirements (e.g., modeling growth under different conditions).

Therefore, the use of systems biology is expected to provide a holistic view of the phenomena involved in resistance, because this approach helps to identify the hub genes with their interaction partners that play a critical role at the molecular level in causing resistance [10]. In this regard, in what follows, we provide a brief history of the development of AR, as well as the mechanisms involved in this event. We also highlight the contribution of omics technologies in advancing the understanding of antibiotic responses at the system level. Importantly, we also address the extension of the methodologies employed in simulations with GEMs, integrating experimental data, to improve our understanding of the involvement of metabolism in the alterations caused by the presence of antibiotics. Future perspectives and limitations in the application of systems biology approaches are also addressed.

## 2. Antibiotic Resistance: A Complex and Multilevel Problem

The occurrence of AR represents a serious and multivariate issue. The World Health Organization (WHO) has warned that the problem of AR must be adequately confronted and contained, because if it is not, then the phenomenon has the potential to cause many problems for the current healthcare system [11]. 

In 1943, acquired resistance of the genus *Staphylococcus* to penicillin was identified, even before the widespread production of this antibiotic, highlighting that these bacteria probably have an intrinsic predilection for resistance stored in their genome, which has been progressing over the years [12]. Therefore, the acquisition of bacterial resistance to antibiotics is a process that can occur naturally and independently of human intervention [13]. However, since the introduction of penicillin as a treatment for bacterial infections, the aptitude of bacteria to acquire such resistance has been accelerated, selecting from naturally resistant or more virulent bacteria, which—combined with the widespread abuse of antibiotics in diverse environments, from humans to animals to agriculture—has rapidly exacerbated this phenomenon.

A fundamental question has a focus on the factors leading to the increase in the phenomenon of antibiotic resistance. Unconscious use of antibiotics is a determining factor in the aggravation of this global problem; however, this is coupled with the poor development of innovative and new antibiotics, which play a key role in the global resistance scenario. The current decline in antibiotic research and discovery is a complex and multifactorial problem. Several of the largest pharmaceutical corporations have currently scaled back or discontinued their antibiotic pipeline divisions due to a lack of cost-effectiveness. Restricted use and shelf life, increasing regulatory fees, generic competition, and inevitable resistance driving greater declines in usage of the drugs caused a risk–reward ratio that is negative (Figure 1). The result in 2013 was that just four multinational pharmaceutical companies still maintained antibiotic development divisions. Consequently, international organizations have enacted programs to encourage companies to continue developing antibiotics [11].

Addressing the multifaceted problem of resistance requires addressing all factors together, from hospital systems to the scientific society. It is critical to understand the mechanisms involved in bacterial resistance to antibiotics, as well as how bacteria acquire this residency, and then use these to seek novel tools with which to understand and perhaps halt the advance of antimicrobial resistance (AR) and its implications for present and future generations.

## 3. Mechanisms of Action and Resistance Acquisition

To understand the need to use techniques that provide a holistic and integrated view of the events involved in antibiotic resistance, it is essential to know the molecular mechanisms by which antibiotics induce cell death or inhibit growth, acting as bactericidal or bacteriostatic drugs, respectively, as well as the genetic plasticity for the acquisition of such resistance. 

In this sense, knowledge of cell death due to the presence of antibiotics has traditionally been associated with different events that may be related to damage to the cell wall. This damage is due to the loss of cell wall structural integrity caused by inhibitors of its synthesis, the arrest of DNA replication as a result of DNA gyrase inhibitors, the arrest of RNA synthesis, or finally, alterations in the translation of proteins promoted by inhibitors of their synthesis.

However, beyond traditional knowledge, many studies have been carried out that highlight the complex events involved in cell death caused by antibiotics, providing evidence to support the theory that lethality could not be strictly excluded from the direct effects of drug–target interaction, as traditionally postulated [2], and support the updating of the concept based on the antibiotic response characterized by the omics studies reported by Brazas and Hancock [14], where more specifically, a range of observational findings in numerous bacterial strains consistently suggest adverse roles for either indirect effects (e.g., activation of stress response and adaptive metabolic responses) or secondary effects (e.g., modification of regulatory and network interactions) in mortality arising due to antibiotic treatment (Figure 2).

Regarding the acquisition of antibiotic resistance, bacteria present an important genetic adaptability that allows them to survive a wide range of environmental factors, even in conditions that jeopardize their existence, such as the presence of antibiotic molecules. For example, it is noteworthy that bacteria coexisting with antimicrobial-producing organisms within common ambient interactions developed evolutionary mechanisms for resisting the effect of the antibiotic molecule, and as a result, they have acquired an intrinsic resistance, allowing them to flourish in the proximity of these organisms [15]. Regarding evolution, bacteria have two main genetic approaches for adapting to the presence of antibiotics: vertical (endogenous) evolution or horizontal (exogenous) evolution [1].

In the work of Laws and colleagues [1], the concept of vertical evolution is described as the development of a spontaneous mutation in the bacterial genome that gives the bacterium (and subsequently its progeny) enhanced resistance to a particular component. The process of gaining such resistance is usually gradual, whereby the selective stress of antibiotic presence leads to a starting mutation that enables the mutant bacterium to survive, with subsequent additional mutations conferring a resistance benefit during subsequent antibiotic treatment. Frequently, mutational changes leading to resistance have large consequences for cellular homeostasis (i.e., a decrease in fitness), and are often not conserved because they are necessary only in the presence of the antibiotic. In general, mutations leading to antimicrobial resistance disrupt antibiotic action through different mechanisms (see below). Therefore, resistance arising from acquired mutational changes is diverse and ranges in complexity [15].

On the other hand, Laws and colleagues [1] provide a comprehensive review of horizontal (exogenous) evolution. According to the authors, the transfer of a gene involved in resistance to another susceptible bacterium is the definition of horizontal evolution. This process represents a major motor of bacterial evolution and is usually the cause of the evolution of antibiotic resistance. Three mechanisms may be involved in this process: conjugation, transduction, and transformation. The first of these implies the transference, between bacteria, of an R-plasmid (resistance-plasmid) carrying AR genes by means of a conjugative pilus. The second process, transformation, involves modification of the bacterial genome due to the insertion of external DNA, and finally, transduction implies the transfer of bacterial DNA provided via a viral vector. Unfortunately, these transfer mechanisms permit a resistance gene, gained by relatively minor problematic bacterial species, to be transferred to a potentially harmful bacterial strain, with potentially damaging effects [1,16].

Overall, AR is a complex process involving different levels, and therefore, an approach involving the development of a comprehensive and predictive understanding of both the mode of action of antibiotics and the physiology of antimicrobial stress is needed, which can be achieved through systems biology, by employing a synergistic integration of omics at the multilevel, providing the framework for contextualizing the genetic and metabolic changes involved to antibiotic resistance-related events.

## 4. Systems Level: The Increasing Use of the Post-Genomic Approach to Understanding Antibiotic Resistance

The first genome to be sequenced was that of *Haemophilus influenzae* in 1995, and since then, the progress in both experimental and computational technologies for the study of biological systems has become enormous [17], leading to the emergence of systems biology concepts [18]. Systems biology investigates the entire biological system arising from individual biomolecules and their interactions [19]. Therefore, this approach has as its paradigm the relationship between genetics and cellular functions in a hierarchical and participatory manner. Cellular functions depend on the joint action of the products of a large variety of genes. This coordination can be thought of as a “genetic circuit”. The term “gene circuit” is designated as a set of various gene products that are jointly necessary to perform a specific cellular role. Individual gene circuits do not operate singly, however, but within other gene circuits. The ensemble of all these circuits performing together within a genome results in cellular functions and drives the hierarchical breakdown of intricate cellular processes (Figure 3). 

Using this concept in the specific case of antibiotic resistance, if a new AR gene would be introduced, or AR mutations would occur in a bacterium, these changes would produce profound effects on bacterial cellular function, altering the expression of many genes and consequently altering the bacterial proteome and its metabolism, having consequences on the entire genetic circuitry. In addition, it should also be considered that depending on the organism, a resistance gene mutation can have different effects.

In this sense, to obtain a complete picture of the mechanisms of AR and the implications of their acquisition for microbial cell function, and in turn, to understand these processes at the level of the genetic circle, systems biology uses specialized bioinformatic tools and approaches capable of analyzing the growing amount of information produced in molecular biology, using—as mentioned above—multiomics technologies, including genomics, transcriptomics, proteomics, and metabolomics. Although the implication of each of the omics technologies mentioned here in relation to AR is discussed below, Table 1 summarizes some of the bioinformatic tools used for the analysis of the different omics technologies.

The primary purpose of the collected information obtained from “-omics” is to obtain simultaneous insights into the occurrence and identification of several thousand genes, in order to examine their genetic linkage to other genes, as well as their expression levels and the proteins encoded by those genes, and finally to measure cellular metabolites [45]. Because the amount of data generated can be enormous, and therefore very complicated to interpret, genomic models (see below) have been developed that are able to collect the enormous quantity of experimentally generated information into mathematical models so as to comprehend it holistically [7,46,47].

In this context, in this part of the review, we highlight recent efforts that have shown that the use of omics technology and systems biology as quantitative and robust hypothesis-generating frameworks can improve the understanding of antibiotic resistance. 

### 4.1. Genomics: First Step to Identify Antibiotic Resistance Genes (ARG) 

For a long time, studies about AR have been restricted to classical microbiology techniques such as culture, isolation, identification of phenotypic characteristics, and finally the determination of the chemical profile of isolates [48]. After identification, the ability to survive at different concentrations of antibiotics could be inferred by determining the minimum inhibitory concentrations [49]. In general, classical culture-based microbiology techniques allow for the characterization of isolates; however, they are limited by speed and scale [50]. Consequently, the characterization of many samples, both environmental and hospital, can become very time-consuming and expensive [50]. Currently, both microbiological research and diagnostic studies combine traditional methods and next-generation sequencing (NGS). Progress in NGS performance associated with the reduced cost of these technologies has enabled whole-genome sequencing (WGS) of a wide variety of microorganisms. WGS analyses are making it possible to identify novel ARGs [51] and study the phylogeny of isolates, in addition to epidemiological traits, which is critical in helping to understand AR [8].

Utilization of WGS to identify novel ARGs has been described in the literature. For example, Grad et al. [52,53] used WGS analysis of *Neisseria gonorrhoeae* bacteria and identified a new *penA* allele in mosaic and *mtrR* mutations in mosaic that confer resistance of this microorganism to cefixime and azithromycin, respectively. On the other hand, Marques et al. [54] sequenced the genome of 17 *Helicobacter pylori* strains from pediatric patients and observed single and combined mutations located in the 23S rRNA gene (A2142C and A2143G), which are linked to resistance to clarithromycin. In another work, Zhu et al. [55] studied the possible horizontal transfer of florfenicol resistance (*floR*) gene-related sequences in *Proteus* strains by WGS. That work revealed that the *Proteus cibarius G11* strain harbored two copies of the *floR* gene: one on the chromosome and the other on a plasmid (pG11-152). In another plasmid (pG11-51), the presence of the chloramphenicol–florfenicol resistance (*cfr*) gene was observed, flanked by two IS26. With this bundle, those authors demonstrated the importance of mobile genetic elements in the replication of the *floR* gene, and in the horizontal transfer of the resistance gene. Along the same lines, Wu et al. [56] sequenced the genome of multidrug-resistant (MDR) *Staphylococcus lentus* strain H29, and found 11 genes conferring resistance to this microorganism, with one copy encoded on the plasmid and the other on the chromosome. Among the genes found, two copies of the mobile genetic element (MGE)-related *floR* genes *cfr* (IS256-cfr) and *fexA* (*radC-tnpABC-hp-fexA*) were included. Finally, Zhang et al. [57] identified, via genome sequencing of tetracycline-resistant *Arthrobacter nicotianae* OTC-16, eight genes related to antibiotic resistance, three of them located nonplasmidically, with obvious mobile features.

As mentioned above, the use of WGS enables the rapid identification of AR-related genes, in addition to providing greater discriminatory ability regarding the genomic epidemiology and antimicrobial resistance determinants of different strains. In this regard, many studies targeting the epidemiology of microorganisms exhibiting resistance genes are being reported in the literature. For example, Boiko et al. [58] elucidated the WGS-based epidemiology and characterized AMR determinants in 150 strains of *N. gonorrhoeae* that spread in Ukraine between the years 2013 and 2018. Overall, those authors found isolates resistant to ciprofloxacin, tetracycline, and benzylpenicillin. The results of phylogenomic analysis highlighted six major groups, the majority of which were associated with the MDR gonococcal lineage. The presence of *GyrA* S91F and *ParC* S87R mutations was associated with resistance to ciprofloxacin; on the other hand, mutations in *rpsJ* V57M and *tetM*, were revealed to be involved in resistance to tetracyclines; the *penA*-34.001 mosaic with penicillin, and finally, the presence of *mtrR*, *PorB1b*, and *G101D* genes; and PBP1 L421Pla mutations with resistance to β-lactamases. 

In line with the above, Rokney et al. [59] also used WGS to study the occurrence and genetic basis of AMR in 263 *Campylobacter jejuni* isolates recovered from a national collection during 2003 and 2012. The results obtained with genome sequencing were compared with experimentally determined phenotypic resistance. The most prevalent resistance-related genes found were *cmeABC* (related to efflux pumps); *tet*(O) (tetracycline resistance gene); *aadE* (streptomycin resistance gene); and finally, a quinolone resistance-point mutation, *gyrA* T861. The highlight of their study is that they detected 12 genes conferring resistance to β-lactams in 241 isolates, with *bla*_OXA-580_, *bla*_OXA-461_, and *bla*_OXA-193_ being the most prevalent. Thus, global correlation rate between WGS-based genotypic prediction and phenotypic resistance was 98.8%. 

In another study by Onofrio et al. [60], 24 clinical isolates of colistin MDR strains (1 *Enterobacter aerogenes*, 8 *Acinetobacter baumannii*, 1 *Enterobacter cloacae*, and 14 *Klebsiella pneumoniae*) were collected from 4 different hospitals located in Croatia during the period from 2013 to 2018. The study aimed to analyze the molecular epidemiology and mechanisms of colistin resistance by WGS of all isolates. It was observed that 12 strains of *K. pneumoniae* were widely resistant to colistin, but on the other hand, they found *bla*_OXA-48_ (carbapenem resistance gene most prevalent in Croatia and other places in Europe) in 63% of the isolates. All *A. baumannii* isolates possessed OXA-23 type oxacillin hydrolyzing carbapenemases, and five were pandrug-resistant. Most likely, colistin resistance was chromosome-mediated. Moreover, previously reported mutations that are associated with colistin resistance were identified (*PmrB, PhoP, PhoQ*, and *MgrB*). In the overall phylogenetic analysis, the DNA mutations causing the MgrB protein mutations were mainly present in the lineages comprising the colistin-resistant strains, and the second most prevalent mutation (K3X) was also found in the strains. 

Furthermore, Lee et al. [61] employed WGS as a screening tool in the surveillance of 1025 bloodstream-infection-associated *Enterococcus faecium* isolates collected across Australia from 2015 to 2017. WGS analysis identified three distinct clusters of isolates with additional subgroups. One cluster harbored mainly non-CC (clonal complex), while others were dominant for the vanA and vanB operons. In addition, different dominant subclusters were observed in each region of Australia. According to the authors, these results may help to place future surveillance data in a broader perspective that includes the detection of new E. faecium strains in Australia and the dissemination and evolution of each strain.

Finally, Butin et al. [62] compared genetic, microbiological, epidemiological, and clinical characteristics among linezolid-resistant *Staphylococcus capitis* (LZR) isolates from ICUs in France and other European countries. The genetic relationship between 21 LZR isolates (1 from Finland, 9 from France, and 11 from Greece) was investigated by WGS. All microorganisms studied were resistant to both aminoglycosides and methicillin. In addition, the authors identified a G2576T mutation in 23S rRNA in every strain (*cfr* and *optrA* genes were absent). WGS analysis identified at most 212 SNPs across the central genomes of the LZR strains. Finally, their efforts identified and characterized an LZR clone of *S. capitis* spread across three different sites in Europe, harboring the common multiple resistance and a G2576T mutation in 23S rRNA.

### 4.2. Emerging Discoveries through Transcriptomics Approach

Transcriptomic analyses highlight overall alterations in gene expression in response to environmental events, providing a snapshot of the genes that act under a given condition. In the case of antibiotic resistance, this picture is the result of variation in gene expression patterns, which provides insight into the mechanism of action of a particular antimicrobial being studied, as well as the general physiological responses of bacteria to stress related to the presence of such molecules. Further understanding of the mechanism of action of known antibiotics could help to reveal new strategies to contain the spread of resistance genes [14]. Thus, in recent years, important advances have been made in efforts to understand AR using this technology.

For example, *Pseudomonas aeruginosa* is known to be one of the main bacteria causing morbidity and mortality in the hospital setting, in addition to being the main cause of nasocomial infections. A recent study reported the resistance of a strain of this microorganism to ceftazidime/avibactam (CZA), ceftolozane/tazobactam (C/T), and piperacillin/tazobactam (P/T). On the one hand, they showed that a single-nucleotide variation, causing the G183D mutation in the β-lactamase *AmpC*, was responsible for resistance to CZA and C/T. On the other hand, resistance to P/T was related to DNA methylation because transcriptomic analysis revealed that methylation involved 14 genes that were differentially regulated. Since upregulation of *opdQ* gene, responsible for the OprD protein (from the porin family), was observed, those authors suggested that *P. aeruginosa* susceptibility to P/T might be due to epigenetic regulation of opdQ expression (Huang et al., 2020). In another study, oprD mutation combined with *ampC* upregulation was shown to contribute to the increased carbapenem resistance in the isolate IRP41 [63]. The efflux pump mechanism related to AR has also been described in the *Pseudomonas aeruginosa* strain [64].

As for *Salmonella*, Li et al. [65] investigated the resistance mechanism of *Salmonella* Typhimurium ATCC13311 (AT) to polymyxin. RNA-seq and RT-qPCR showed increased expression of the two-component system (*phoP*, *phoQ*), and consequently, loss of the *phoP* gene in the AT-P128ΔphoP mutant decreased polymyxin resistance by 32-fold. This fact suggests that the two-component *phoPQ* system performs a critical role in polymyxin resistance in *Salmonella*. In another work, Zhang et al. [57] elucidated the molecular mechanisms behind the mode of action of *CpxAR*, and efflux pumps were found to synergistically enhance the antibiotic susceptibility of *S. typhimurium* to colistin. Thus, deletion of *cpxR* and *tolC* causes significantly increased susceptibility of S. typhimurium to colistin. In another study, it was also shown by RNA-seq that loss of the *AcrB* protein, related to the *AcrAB-TolC* operon (multiple drug resistance efflux (RND) system in Gram-negative bacteria), causes loss of virulence in *Salmonella enterica serovar* Typhimurium [66]. Nghiem et al. [67] examined the resistome characteristics of *Salmonella spp*. from different pollution samples collected in Hanoi, Vietnam. RNA-seq analysis showed the presence of 107 overexpressed ARG, among which 8 related to quinones, 22 to β-lactams, 7 to chloramphenicol, 46 to aminoglycosides, and 6 to tetracyclines, as well as 6 sulfamide-trimethoprim resistance genes, and finally another 12 genes related to undefined antimicrobial resistance. Moreover, mutations in the *parC* gene (S80R and the new A628S mutations) and in *gyrA* (S83F and D87G) have also been shown to contribute to ciprofloxacin resistance in *Salmonella indiana*. Finally, Gu et al. [68], by analyzing transcriptomic profiles of *Salmonella enterica serovar Enteritidis* (*S. Enteritidis*), demonstrated that *purBCDFHKLMNT* were the core genes for fluoroquinolone (CF) resistance.

In *Staphylococcus aureus*, a comprehensive RNA-seq analysis to investigate the gene expression profile after exposure of different strains to vancomycin revealed 99 overexpressed genes related to antibiotic resistance, which were then compiled to create a multiresistance of 25 known and novel genes identified as playing an important role in antibiotic resistance. Among the known genes, *agr* and other virulence factors were highlighted, that together help in activating the detection of quorum-sensing (*qs*) systems [69]. Recently, Wang et al. [70] studied a series of compounds with strong antimicrobial activity, obtained from endotype B polycyclic acylphloroglucinols (PPAP). One of the derivatives, PPAP 23, was selected for screening of its bactericidal characteristics and mode of action. The antimicrobial mechanism of PPAP 23 was investigated by RNA-seq in methicillin-resistant *Staphylococcus aureus* (MRSA). The results of this technique demonstrate that PPAP 23 indicated iron depletion in bacterial cells, since genes implicated in iron transport appeared to be downregulated and the iron storage gene was upregulated.

Recently, transcriptional studies have also been performed on *Escherichia coli*. Cho et al. [71] characterized compensatory mutations acquired from the lack of the major antibiotic efflux pumps *AcrEF* and *AcrAB* with mapping in four regulatory genes (*rpoB*, *baeS*, *hns*, and *crp*). These mutations may activate alternative pathways, thus increasing antibiotic resistance. Results obtained with transcriptome sequencing (RNA-seq) indicated that DNA and protein biosynthesis pathways were downregulated, while pathways to combat various stresses were upregulated. Those authors highlighted that compensatory mutations may interact synergistically to promote AR to a degree comparable to that of the efflux pump-competent parental strain.

Resistance of *A. baumannii* is a major barrier to the treatment of clinically relevant infection. In this regard, Alkasir et al. [72] cultured *A. baumannii* ATCC 19606, which was grown at 50 times the minimum inhibitory concentration of certain antibiotics, and the transcriptome of cells grown in this condition (persister cells) was carried out. The data obtained showed that there was a significant increase in two toxin/antitoxin systems: GP49 (*HigB*)/*Cro* (*HigA*) and DUF1044/*RelB*. In addition, cultivation of *A. baummannii* in a persister condition altered the metabolism of this microorganism, which had genes related to tricarboxylic acid (TCA) cycle, electron transport, and adenosine triphosphate [ATP] downregulated during growth in this condition, while genes for degradation of aromatic compounds were upregulated. These findings strongly support the involvement of aromatic compound degradation genes in persister growth and maintenance. In another work, the efflux pumps, *craA*, involved in chloramphenicol resistance were elucidated using the transcriptomic approach [73]. The influence of environmental conditions, in the case of nutrient restriction in downregulating the expression of the two promoters of the *craA* gene, was seen. This fact shows that downregulation of *craA* gene regulators conditions the cells to be more sensitive to chloramphenicol.

### 4.3. Recent Trends in Proteomics

Although transcriptomics is the most widely used technique with which to investigate bacterial responses to antibiotic stress, proteomics has been widely employed to progress the comprehension of the steady-state proteome of antibiotic-resistant clinical isolates [74]. In recent years, many researchers have focused their efforts on identifying and designing protein biomarkers related to susceptibility or resistance. These advances are being made possible by improved technologies applied to proteomics, such as matrix-assisted laser desorption ionization time-of-flight mass spectrometry (MALDI-TOF MS). 

Multiple methods based on MALDI-TOF MS for rapid detection of antimicrobial resistance have been proposed. In this context, an early application of MALDI-TOF MS related to AR was the observation of the decline of the peak related to the β-lactam class antibiotic, and the apparition of peaks corresponding to their hydrolysis products upon the exposure of β-lactamase-producing bacteria (aerobic and anaerobic) to those β-lactam antibiotics [75]. In another study on the application of this methodology, Singh et al. [76] reported the determination of 27 proteins in clinical isolates of M. tuberculosis whose abundance was specifically increased in MDR isolates.

Although MALDI-TOF MS-based methods have been used successfully for ARG detection, there has been a growing trend toward label-free quantification using spectral counts or chromatogram peak areas, such as liquid chromatography–tandem mass spectrometry (LC-MS2) analysis. For example, Uddin et al. [77] used this approach to compare the proteomic profile of three important pathogens resistant to different antibiotics: laboratory-derived and clinically isolated *S. typhimurium*, *K. pneumonia*, and *S. aureus*. Those authors highlight that the most significant finding of their study is that not only were proteins related to AR identified, but also other bacterial membrane proteins that were not initially associated with the development of AR in these bacteria. In another similar example of the use of this methodology in studies related to antibiotic resistance, Kittisenachai et al. [78] evaluated the prevalence of clarithromycin and metronidazole resistance in the pathogen *H. pylori*, as well as dual resistance to both antibiotics. The authors suggested a link of rpoB to metronidazole sensitivity and of FBPAII to sensitivity to the other antibiotics studied. 

The use of proteomics allows for a more holistic molecular view of AR in comparison to conventional methods; however, in many cases, the use of these approaches is associated with other types of omics, and the association with WGS is observable in several studies. For example, Foudraine et al. [79] performed a systematic analysis of resistance to different antibiotics (meropenem, third-generation cephalosporins, aminoglycosides, and ciprofloxacin) in 187 isolates of *K. pneumoniae* and *E. coli* that harbored different AR mechanisms. Those authors showed that proteins of different antimicrobial resistance mechanisms can be screened by a proteogenomic analysis using a bottom-up LC-MS2 proteogenomic approach. The authors also stated that although not all ARG mechanisms were determined at the protein level, resistance could be accounted for by the proteins identified in most of the isolates studied. Similarly, Li et al. [80] used liquid chromatography–tandem mass spectrometry (LC-MS2) coupled to metabolomics to systematically compare the profiles of a mutant in a maltose-specific channel porin (*ΔlamB*) and their wild-type strain of *E. coli*, with and without the presence of ciprofloxacin (CFLX). Their efforts demonstrated that suppression of *lamB* in the presence of the antibiotic resulted in downregulation to a variety of important metabolic pathways. Many proteins related to pyrimidine metabolism, as well as amino acid-tRNA biosynthesis, were not modified in the *ΔlamB* strain; however, they were decreased in the presence of CFLX. Those authors highlighted a downregulation of *lamB*-modifying intracellular metabolism, leading to an increase in bacterial resistance to antibiotics. 

### 4.4. Accelerated Growth in the Use of Metabolomics

Bacterial metabolism carries a significant role in AR and may contribute to its acquisition or change due to exposure to these antimicrobials [81]. For example, high metabolic activity is essential to favor resistance events, as it is necessary to activate a broad molecular machinery, such as cell wall modifications, mutation stabilization, transport, energy generation, and overexpression of efflux pumps (Figure 4). In this sense, understanding the metabolic processes involved in this mechanism could help to strategically modify bacterial metabolism with the aim of resensitizing it to treatment.

Through approaches such as metabolomics, it is possible to obtain a global view of all the molecules implicated in metabolism, which in turn is intrinsically connected to the phenotype of the organism. Knowledge of this snapshot of the molecules involved in metabolic processes reveals the early responses to antibiotic stress and the adaptations necessary to maintain AR mechanisms. Metabolomics approaches are indispensable tools for understanding the relationships between the mechanisms involved in AR and bacterial metabolism.

As with other omics technologies, metabolomics can be employed with different methodologies and approaches. In general, the most widely employed approaches for metabolite detection are mass spectrometry (MS) and nuclear magnetic resonance (NMR) spectroscopy, with a huge variety of combinations of analytical tools that can be combined with mass spectrometry. In terms of approaches, these can be differentiated generally into untargeted metabolite profiling and guided methods. Nontargeted methods aim for a broad coverage of metabolites but may not allow for complete identification of molecular structures. Targeted metabolomic methods have the goal of quantitative analysis of a metabolite pool, with a higher structural resolution of the selected metabolites identified.

Regarding including targeted metabolomics in the topic of antibiotic resistance, a wide range of studies have employed these approaches associated with mass spectrometry and coupled to different analytical tools [82,83]. Although mass spectrometry is the most widely used methodology in the study of resistance, some studies have used NMR [84]. In a recent report in which the targeted metabolomics approach was applied to antibiotic resistance, Schelli et al. [85] studied the metabolic alterations caused in two isogenic strains of *S. aureus*, with different susceptibility to methicillin, in the presence of norfloxacin, kanamycin, and ampicillin. A Thermo Scientific Ultimate 3000 HPLC coupled with a TSQ Quantiva Triple Quadrupole mass spectrometer was used in this study. A hydrophilic interaction chromatography (HILIC) column was purchased from Waters Corporation (Milford, MA, USA). The authors reported that depending on the presence or absence of methicillin, there were more metabolic variations between the two strains in the different antibiotics, especially in the metabolism of amino acids, pyrimidines, and purines. In addition, the authors observed metabolic differences between the two isolates in the presence of the same antibiotic, suggesting that the susceptible and resistant strains have a different stress response mechanism.

In contrast, nontargeted metabolomics allows us to explore and characterize a wide variety of metabolites. The use of this approach allows us to obtain a comparatively fast metabolic fingerprint, which can be employed to detect metabolic adjustment upon ARG development under a broader range of different conditions with a larger throughput [81]. Because of this feature, this approach has been widely used in the field of AR [86,87,88]. An example is the recently reported work by Han et al. [89], where they used untargeted metabolomics to provide insight into the molecular mechanisms behind *P. aeruginosa*-related polymyxin resistance. To do so, the authors used LC/MS to compare the metabolite changes that occurred in two *Pseudomonas aeruginosa* strains—one polymyxin-susceptible (PAK) and one resistant (PAKpmrB6)—when exposed to polymyxin B. A total of 1297 metabolites were identified, and the most significant metabolic changes occurred after 1 h of exposure to the antibiotic studied. In both strains studied, polymyxin produced osmotic stress, as reflected by an elevation of the trehalose-6-phosphate level. In addition, the polymyxin-susceptible strain revealed a considerable reduction in lipopolysaccharide and peptidoglycan synthesis. The authors claim that these results could be used in the further development of a next generation of polypeptide antibiotics [89]. In another study, using the same approach, Zampiere et al. [90], demonstrated that early metabolic changes observed in *E. coli* after treatment exposure to a wide variety of antibiotics may reflect the mechanisms of drug action and reveal the relationship between metabolism and its role in the primary stress response to antimicrobials. Furthermore, the authors suggest that modification of bacterial metabolism may be a strategy to disrupt the primary response to antibiotic treatment, and thereby decrease the likelihood of survival and subsequent progression of antibiotic resistance. 

Notwithstanding the two approaches mentioned above, targeted and nontargeted metabolomics are widely used to infer metabolic changes under stress conditions, such as antibiotic exposure. Both have their advantages and disadvantages, so the decision as to which approach should be employed will depend on the objectives of the study. In this regard, the use of targeted metabolomics has the disadvantage of requiring prior knowledge of the metabolites to be determined [81]. Another limitation of this approach is the coverage of the metabolites determined. Since it is necessary to define the metabolites to be identified in advance, it is possible that some metabolites that are important for the process being studied may be missed. On the other hand, the targeted approach has the advantage that the data obtained are easier to handle, in addition to being an excellent strategy for the determination of specific processes. As for nontargeted metabolomics, this approach allows for the determination of unusual metabolites, such as in the case of chemical processing of antimicrobials, or if the range of metabolites is complex to identify. This approach is suitable for the determination of novel metabolic pathways and in the case of not knowing in depth the metabolic processes involved in the exposure of microorganisms to environmental stress conditions. 

Considering these limitations, both approaches are suitable to be used in studies related to antimicrobial resistance, but experimental planning is necessary when deciding which of the two approaches is the most suitable to obtain the desired answers about the involvement of antimicrobials in bacterial metabolism.

### 4.5. Metabolic Models to Expand the Comprehension of Mechanisms Associated with Antibiotic Resistance

It is known that the development of resistance affects bacterial metabolism, mainly altering growth. This is due to the high energetic cost necessary to maintain the events involved in resistance, but it is not only that, because often the expression of several genes, which are sometimes not associated to the acquisition of the resistance phenotype, is also altered. These alterations could provide many clues about the effects and mechanisms involved in resistance.

As mentioned above, the use of different omics approaches has provided a large and valuable amount of information on genetic modifications and gene and protein expression, as well as metabolite variations in the field of antibiotic resistance. However, the information obtained through these approaches is disconnected, and integration with GEMs allows us to obtain a holistic view of the events associated with antibiotic resistance. In this sense, GEMs interconnect genes, enzymes, proteins, metabolites, and finally metabolic reactions. Further, these models can make predictions of possible metabolic responses by coupling transcriptional responses with phenotypes, a very important feature for understanding antibiotic resistance.

With the sequencing of the first whole-genome sequences during the mid-1990s [17], in principle, the possibility of identifying every gene product implicated in complex biological functions in a wide variety of organisms became feasible. Biochemical knowledge of metabolic processes allowed for the reconstruction, at the genome scale, of the metabolic networks of a particular organism in a biochemically complex manner [91,92]. These metabolic networks, or GEMs, are currently the only approach that allows the metabolism of an organism to be modeled and analyzed through a global analysis [93]. The construction of GEMs is tedious work. Thiele and Palsson [94] defined 96 steps to obtain a high-quality metabolic model; however, these can be summarized in four principal steps—draft reconstruction, manual curation, mathematical model conversion, and network analysis (Figure 5).

The first step is to obtain a draft reconstruction that is supported by the target organism genome annotation and different biochemical databases. This first draft is usually obtained in an automated way, and there are a variety of databases that allow one to download models based on genome annotation; therefore, it is a compilation of metabolic reactions encoded in the genome. As these are automatic annotations, there may be errors such as incomplete, erroneous, and missing annotations. The second step of the process is based on the trimming and refinement of the components of the model. Metabolic functions and reactions contained in the preliminary reconstruction draft are tested separately against the organism-specific literature. Inclusion of non-organism-specific reactions potentially impacts the performance of the resulting models in terms of predictive behavior. In addition, during this stage, detailed data about biomass components, maintaining parameters, and growth conditions are gathered, serving as input for the simulations. In the third stage, the initial reconstruction is represented in SBML (Systems Biology Markup Language) format, and the simulation conditions are defined, i.e., the input and output. In this conversion, a stoichiometric matrix (S) is generated that corresponds to the connection between the metabolites (rows) and the metabolic reactions (columns) involved in the reconstruction. The numerical values that compose this matrix correspond to the stoichiometry of the consumption and/or production of a metabolite in each reaction, i.e., a positive number represents the production of the metabolite, while a negative number represents its consumption. The mathematical matrix is the basis of the metabolic models, in which the conservation of masses is considered, and for its resolution, the equation (*S.v = b*) is considered, where *S* is the matrix, *b* is the accumulation of the metabolite, and *v* is the vector of reaction fluxes. When using, for example, flux balance analysis (FBA) to solve the system of differential equations, it is essential that the organism is in a stable state of growth (steady state) so that *b* = 0, resulting in a linear system of equations [95]. Thus, the system can present a large solution space, and that is why the fluxes are limited to using upper and lower constraints on each reaction individually (*v_i_*): *v_i lower_* ≤ *v_i_* ≤ *v_i upper_*. These bounds allow the reconstruction to be used to simulate specific conditions. It is noteworthy that the model generated initially may differ, both in scope and limits, from the model obtained at the end, which is due to the multiple validation and refinement performed to obtain a robust model capable of simulating the phenotypic behavior with a high degree of correspondence with that obtained in vivo [96]. The fourth and final step of the process of obtaining the reconstruction consists of the evaluation followed by the validation of the network. The final model generated in the third step is systematically evaluated, checking, among other things, its capacity for “growth”, i.e., for its ability to synthesize compounds such as amino acids, nucleotide triphosphates, and lipids, which are the precursors of biomass. This assessment often results in the determination of the remaining metabolic functions missing in the reconstruction, termed gaps in the network, and these are inserted by repeating steps 2 and 3 in part.

Once a GEM has been obtained, by using different approaches these models can help predict cell phenotypes under different environmental conditions, such as the presence of antibiotics. There are a variety of computational techniques that are used to elucidate metabolic features and help to obtain a global view of the processes under study. A constraint-based analysis technique, flux balance analysis (FBA), already mentioned above, is fundamental to understand the metabolic pathways used by a given organism under different environmental conditions. This technique is also used to observe if the deletion of one or more genes can stop the targeted function in bacterial metabolism. Another widely employed technique for the study of metabolism using GEMs is flux variability analysis (FVA), which determines the range of fluxes in alternative pathways capable of reaching the identical target, and can be employed for the identification of potential drug targets [97]. This flux sampling calculates every solution with statistical significance when the target is unclear [7]. On the other hand, GEMs are also very powerful for gene deletion screening studies, as this approach to computational analysis, carried out in just fractions of a second, saves substantial amounts of time and labor compared to conventional laboratory analysis [45]. Finally, a great advantage of GEMs is the ability to integrate with different omics technologies, and the use of this approach allows us to analyze and predict changes in transcriptional regulation effects on cellular metabolism at the systemic level. Knowing that the transcriptional response related to antibiotics is not stress-related and does not confer any fitness advantage, the use of this computational approach hence permits a topological analysis of the networks, thus allowing us to design a more realistic high-throughput experiments [98].

In this regard, several studies have been published that reveal applications of GEMs in antibiotic resistance. Recently, the metabolic pathways involved in polymyxin resistance in *P. aeruginosa* were analyzed using the iPAO1 metabolic model and transcriptomic data [99]. First, FBA was carried out using the experimental data related to lipid A modifications as a constraint, and then the metabolic model was integrated with transcriptomic data. The growth and metabolism of *P. aeruginosa* were slightly affected in simulations using lipid A modifications as a constraint; however, the physicochemical properties of the outer membrane were significantly affected. In addition, simulations performed with transcriptomic constraints revealed a wide variety of metabolic signatures in response to polymyxin treatment, including decreased biomass biosynthesis, enhanced amino acid catabolism, enhanced flux across the tricarboxylic acid cycle, and augmented redox exchange. In another study, FBA was carried out to verify the metabolic changes and flux distribution variation in streptomycin- and chloramphenicol-resistant strains of *Chromobacterium violaceum* using the iDB858 metabolic model and metabolomics data [100]. FBA was performed to predict metabolic alterations due to stresses generated by the presence of antibiotics. For this purpose, experimental constraints were used to represent susceptible and resistant populations of the studied antibiotics. Variation in the distribution of fluxes in response to the presence of the two antibiotics studied was observed. The presence of chloramphenicol increased the overflow of acetate and formate, and this change is associated with fermentative metabolism, through an excess of reducing equivalents and an increase in the NADH/NAD ratio. On the other hand, the presence of streptomycin increased acetate production. Furthermore, we predicted a reduction in proton gradients and a change in proton motive force (PMF) induced by the presence of both antibiotics, and verified the predicted results by experimentally verifying them with flow cytometry-based membrane potential measurements. 

Recently, adaptive laboratory evolution (ALE) experiments have been successfully employed to study the evolution of AR in controlled environments. Within this approach, GEMs have been used to contextualize the genetic and metabolic changes involved in resistance development and studied using ALE. Metabolic adaptation accompanied by the development of AR was studied in the opportunistic pathogen *Pseudomonas aeruginosa* using the UCBPP-PA14 model [101,102]. The impact of deletion of different genes in 42 different carbon sources was contextualized using single-gene knockout simulation, and the results were compared with experimental data [102]. That study highlighted those deletions of the *gnyABDHL* group genes in the evolved piperacillin lineage and resulted in the loss of L-leucine utilization. These findings emphasize the interconnectivity of AR and metabolism and support future efforts to consider this relationship in the design of new antibiotic regimens [101]. Laboratory-controlled evolutions were also established in chloramphenicol-resistant pathogens and streptomycin-resistant *Chromobacterium*. The iDB149 metabolic model was used to predict the metabolic basis of antibiotic susceptibility and resistance. The model predicted electron imbalance and skewed NAD/NADH ratios due to the presence of the antibiotics studied: chloramphenicol and streptomycin. The resistant pathogen reconfigured its metabolic networks to compensate for the altered redox homeostasis [103].

## 5. Limitation of the Use of Systems Biology

Despite numerous applications and early successes, systems biology faces numerous challenges and certain limitations, even in the field of omics technologies, as well as in modeling using GEMs. With respect to omics technologies, sequencing-based approaches such as WGS cannot yet identify unknown resistance mechanisms that may contribute to phenotyping. Another drawback of these approaches is that they often require the isolation of the microorganism. To avoid this problem, a variety of technologies based on the analysis of the microbial community at different levels, such as metagenomics, metatranscriptomics, metaproteomics, and metametabolomics, would be an option in the case of isolated microorganisms. As an example, the use of the metagenomic approach using techniques such as Nanopore, which is portable, fast, and provides long sequence reads, can be implemented at least in microbiomes of low complexity, such as those involved in orthopedic implant infections [104].

Another possible limitation of systems biology comes because of one of its main premises: that of combining large-scale multiomics data to obtain a more holistic understanding. Dealing with the large amount of data generated using different omics technologies is very complex and requires extensive knowledge of bioinformatics to analyze it. This is not necessarily a drawback if the teams include bioinformaticians, but rapid translation of the results to the clinic requires the development of user-friendly tools, which are not always available. In this context, mention should also be made of the lack of suitable databases and analytical tools, such as visualization tools, and those that are available often require prior knowledge of programming.

Concerning GEMs, one of the main shortcomings of this tool lies in the calibration of these models. On the other hand, the lack of kinetic information from GEMs [105], represents a challenge for the integration of omics data, being very important to consider the best way to represent omics data so that they can be integrated into GEMs. Furthermore, modeling based solely on genome and/or transcriptome data may be limited because approximately 50% of the changes found in the transcriptome may not be present in the proteome, and an even smaller percentage of changes in the genome may leak into the proteome.

Finally, several questions have been raised about the true utility of the multiomics approach, as it has so far provided hardly any groundbreaking results or significant mechanistic insights. Therefore, overcoming these challenges is the way forward for this burgeoning discipline. It is also important to remember that future efforts will have to creatively address the major open questions about how to integrate metabolic models with other layers of biological complexity and their associated uncertainties.

## 6. Concluding Remarks

The emergence of multiresistance to antibiotics in different bacterial species is a matter of great concern for health authorities worldwide, to the point that the situation could be comparable to that of the pre-antibiotic era in 30 years’ time. Consequently, the 21st century is facing a new challenge that goes beyond biomedicine and whose solution does not involve a single agent. On the one hand, healthcare systems, medical professionals, pharmacists, and other citizens must be aware of the rational use of antibiotics, particularly because of the main aggravating factor, and the excessive use of antibiotics in humans and animals, which is the main contributor to the development of resistance in bacteria. On the other hand, researchers must strive to understand the mechanisms related to the acquisition of antibiotic resistance.

Traditional studies related to AR have long been based on the search for specific genes and/or mutations that might confer resistance to a given microorganism. Although this approach is relevant, it is not sufficient to provide a holistic view of the elements involved in the emergence and spread of AR, as it is a complex process that occurs at the system level. For this reason, the use of omics technologies has gained ground, and they have been widely used in resistance-related studies, as well as proved to be valuable tools in AR studies.

The possible implications of metabolism in the acquisition and maintenance of resistance, as well as their ability to be integrated with data obtained by omics technologies, has encouraged the growing use of genome-scale metabolic models in resistance-related studies. The major limitation in their use as tools is that they require some bioinformatics training on the part of the users, in addition to the time-consuming and laborious process of obtaining the models.

The outlook for the advancement of knowledge of complex biological systems will depend largely on advances in the integration of high-performance analytical methods (such as omics) and powerful synthesis tools (such as natural multi-hierarchical computation, GEMs, Big Data, or artificial intelligence). In the case of AR, detailed and global genomic information should become increasingly available, feeding integrative models to extract prevailing trends and associate them with anthropogenic interventions. Although systems biology has its limitations, the information provided by using this approach will be an important step forward in establishing rational, ecological and evolutionary approaches to address this important health problem.

## Figures and Tables

**Figure 1 microorganisms-10-02362-f001:**
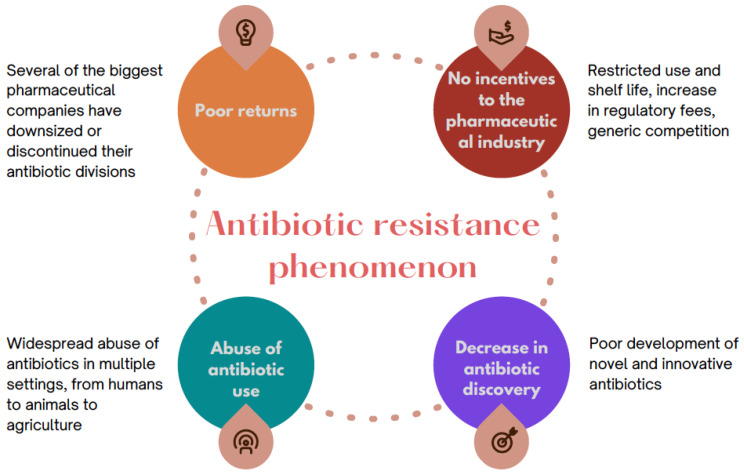
Conceptual illustration of the multivariate issues involved in the phenomenon of antibiotic resistance.

**Figure 2 microorganisms-10-02362-f002:**
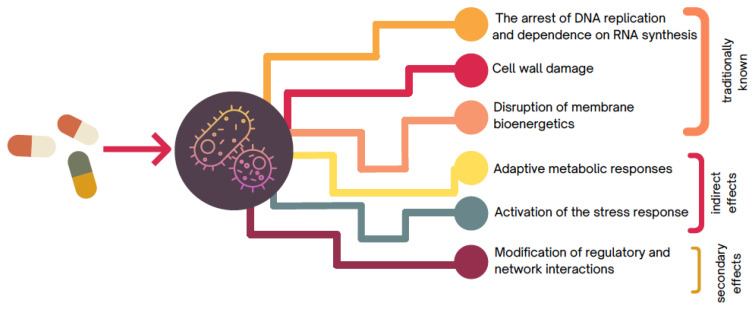
Illustration of the different events involved in cell death caused by antibiotics.

**Figure 3 microorganisms-10-02362-f003:**
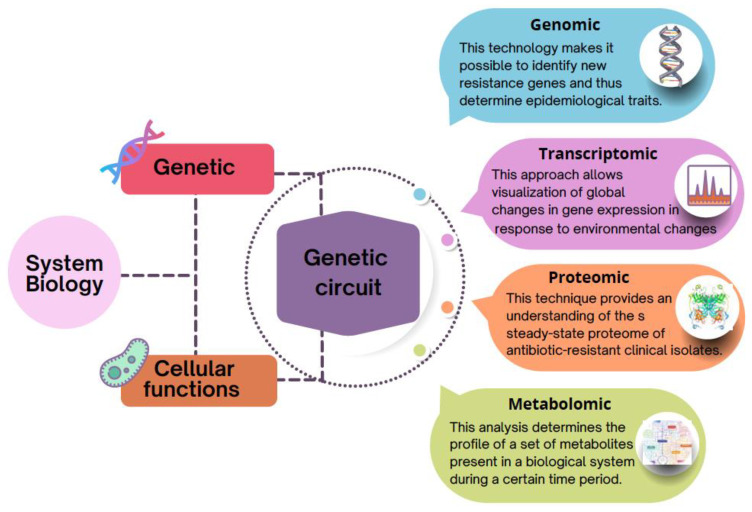
Schematic representation of the genetic circuitry that interconnects the relationship between genetics and cellular functions in a multilayered hierarchy.

**Figure 4 microorganisms-10-02362-f004:**
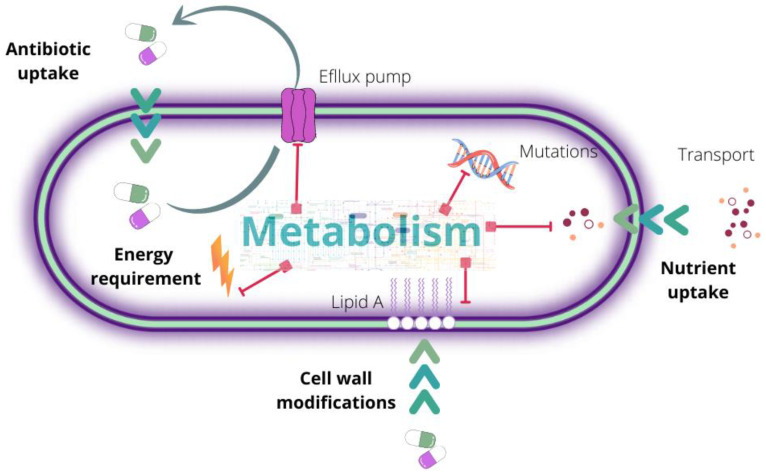
Illustration of the involvement of metabolism in the acquisition of bacterial resistance.

**Figure 5 microorganisms-10-02362-f005:**
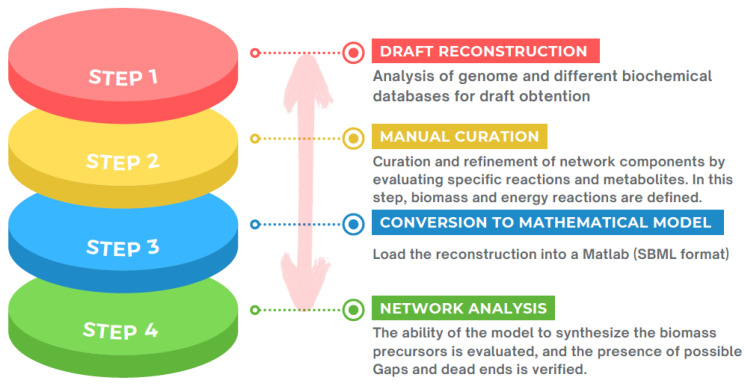
Schematic representation of the workflow for obtaining high-quality GEMs.

**Table 1 microorganisms-10-02362-t001:** Bioinformatics tools.

	Tool	Reference
	ARG-ANNOT	[20]
	CARD/RGI	[21]
ARGs	AMRFinder	[22]
	ResFinder	[23]
	PointFinder	[24]
	Fastp	[25]
Preprocessing and assembly-WGS	SPAdes/	[26]
	Flye	[27]
	Mzmine3	[28]
	MetaboAnalyst 5.0	[29]
Metabolomic analysis	MetFlow	[30]
	Omicsnet	[31]
	PaintOmics 3	[32]
	DESeq	[33]
	edgeR	[34]
General tools for transcriptomic analysis	limma	[35]
	HTseq	[36]
	Rcount	[37]
	Cufflinks-Cuffdiff	[38]
	BioCyc	[39]
	BioMet ToolBox 2.0	[40]
GEM reconstruction	Kegg	[41]
	GeneOntology	[42]
	Pathway-tools	[43]
GEM analysis	CobraToolbox	[44]

## Data Availability

Not applicable.

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
