# Peer review of "Systems Biology: New Insight into Antibiotic Resistance"

_microorganisms, 2022, doi:10.3390/microorganisms10122362_

Round 1

Reviewer 1 Report

The present review provides an interesting perspective of antimicrobial resistance from the angle of multiomics and systems biology.The review is overall well-written and easy to follow. Few minor suggestions below:

Line 21: In the keywords, replace ‘approache’ to ‘approaches’

Line 60: genome-scale metabolic models (GEMs) was defined in line 39.

Line 82: Can it be double-checked if staphylococcal goes in italics?

Figure 3: The metabolomics portion has ‘in a’ twice. Please modify.

Line 240: H. pylori complete name is not shown prior this. Please modify to Helicobacter pylori

Line 434: lamB in italics?

Figure 5. Both illustrations seem to be redundant. Since the information on the left side is more comprehensive, the right-side illustration seems to not add more information, and can be removed.

Author Response

Response to Reviewer 1 Comments

Point 1. Line 21: In the keywords, replace ‘approach’ to ‘approaches’

The word has been modified

Point 2. Line 60: genome-scale metabolic models (GEMs) were defined in line 39.

 All text has been modified where the genome-scale model was used.

Point 3. Line 82: Can it be double-checked if staphylococcal goes in italics?

The text has been changed and the genus Staphylococcus has been included, which should be written in italic.

Point 4. Figure 3: The metabolomics portion has ‘in a’ twice. Please modify.

 The figure has been modified

Point 5. Line 240: H. pylori complete name is not shown prior this. Please modify to Helicobacter pylori

 The name of the microorganism has been modified

Point 6. Line 434: lamB in italics?

The reviewer is right, lamB must be in italic. All the expression has been modified.

Point 7. Figure 5. Both illustrations seem to be redundant. Since the information on the left side is more comprehensive, the right-side illustration seems to not add more information, and can be removed.

The reviewer is right in his/her appreciation. The figure has been modified

Reviewer 2 Report

Over the past few decades, antimicrobial resistance has emerged as an important threat to public health, resulting from the global propagation of multidrug-resistant strains of various bacterial species. Knowledge of the intrinsic factors leading to this resistance is necessary to overcome these new strains. In this manuscript, the author summarized the application of system biology in improving the understanding of antibiotic resistance. Overall, the current manuscript contains valuable information for publication, but writing part need to be checked more carefully. Here are some specific comments.

1)    What are the limitations of systems biology in the study of antimicrobial resistance? The author did not mention it too much. I think this part needs more in-depth discussion.

2)    It is suggested to put the contents of part 5-9 in part 4 to make the article more organized.

3)    The acronym ‘AR’ for antibiotic resistance appears on line 78. In fact, it should be placed where it first appeared.

4)    The resolution of figure 1 and 3 is too low. The author should improve their resolution to make the content on the pictures clearer.

5)    As a review article, the proportion of references in recent three years is relatively small. It is suggested to supplement more relevant research in recent three years.

Author Response

Response to Reviewer 2 Comments

Point 1: What are the limitations of systems biology in the study of antimicrobial resistance? The author did not mention it too much. I think this part needs more in-depth discussion.

Thanks for the suggestion. a section on the limitations of the use of systems biology has been included in the manuscript.

Point 2: It is suggested to put the contents of part 5-9 in part 4 to make the article more organized.

The work has been reorganized and the contents from 5 to 9 included in section 4.

Point 3: The acronym ‘AR’ for antibiotic resistance appears on line 78. In fact, it should be placed where it first appeared.

Whenever the expression antibiotic resistance has appeared in the text, it has been replaced by AR.

Point 4: The resolution of figure 1 and 3 is too low. The author should improve their resolution to make the content on the pictures clearer.

Both figures have been modified and the quality has been increased.

 Point 5: As a review article, the proportion of references in recent three years is relatively small. It is suggested to supplement more relevant research in recent three years.

Many references have been included and the work now includes more research from the last three years.

Reviewer 3 Report

The review-manuscript is titled " Systems Biology: New insight into Antibiotic Resistance" outlined recent developments in utilizing omics and systems biology to understand antibiotic resistance (AR).

The Introduction states the objectives of the work and provides an adequate background. The omics sections are well written. Overall, a very informative review that covers most of the omics implementations in AR studies. However, this manuscript requires minor revision in certain sections.

1.     Add a table listing tools which can be implemented for these omics’ studies.

2.     Given the limitations in the genome and functional annotations, and the occurrence of different microbes in an environment (either in a living host or physical host), it will be helpful to add a paragraph of limitations or a moving forward section with a brief description of challenges with existing approaches and possible use of meta-genomics, meta-transcriptomics, meta-proteomics, and meta-metabolomics.

Author Response

Response to Reviewer 3 Comments

Point 1: Add a table listing tools which can be implemented for these omics’ studies.

Thank you very much for your comments. A table with the main bioinformatics tools used for the analysis of omics technologies has been added to the work.

Point 2: Given the limitations in the genome and functional annotations, and the occurrence of different microbes in an environment (either in a living host or physical host), it will be helpful to add a paragraph of limitations or a moving forward section with a brief description of challenges with existing approaches and possible use of meta-genomics, meta-transcriptomics, meta-proteomics, and meta-metabolomics.

At the end of the manuscript, a section is included where the main limitations related to the use of systems biology are discussed. Although not much, the use of meta-genomics, meta-transcriptomics, meta-proteomics, and meta-metabolomics is also discussed.

Reviewer 4 Report

The manuscript has discussed an important topic concerned with antimicrobial resistance and has been well structured and written in good language, however, there are some points that the authors should address:

1.       line 130 “, as traditionally postulated 2” please revise

2.       LINE 237:delet “bacteria”

3.       line 240: Helicobacter pylori (please write in full at the first mention)

4.       line 245& 250: strain (not italicized)

5.       line 250: multidrug-resistant (MDR)

6.       line 263: N. gonorrhoeae

7.       line 267: MDR gonococcal lineage

8.       line 279 “bla OXA-580, bla OXA-461, and bla OXA-193”: the gene should be italicized and “OXA-461” should be lower case . This point should be revised through the manuscript.

9.       line 282:MDR strains

10.   line 282: please write the bacterial species in full at the first mention

11.   LINE 285: delete (whole genome sequencing) and write only the abbreviation (WGS) as it was mentioned in the previous sections.

12.   line 286: Klebsiella pneumoniae

13.   line 298-290: and five were pandrug resistant

14.   line 290:an updated reference should be added (Moreover, the first emergence of colistin mcr-10 and fosfomycin fosA5 resistance genes in K.pneumoniae isolated from bovine milk in Egypt was identified using WGS analysis “https://doi.org/10.3389/fmicb.2021.770813”)

15.   line 299: replace “WGC” with “WGS”

16.   line 326: OprD “should be italic.”

17.   line 330-331& 337&341&350” Salmonella typhimurium”: the word “Typhimurium” should start with a capital letter and not be italicized.

18.   line 381 “TCA”: write in full at the first mention

19.   line 405: MDR isolates

20.   line 411: S. Typhimurium, K. pneumoniae, S. aureus

21.   line 490& 648: P. aeruginosa

22.   line 499: E. coli

23.   References:

          All bacterial species should be italicized.

         Reference 13& 14: write the pages and volume

Author Response

Response to Reviewer 4 Comments

Thank you very much for your comments, they have been very important for the improvement of the manuscript, especially in the nomenclature. ALL the suggested changes have been carried out and are marked in red in the manuscript. 

  1. line 130 “, as traditionally postulated 2” please revise
  2. LINE 237:delet “bacteria” 

3. line 240: Helicobacter pylori (please write in full at the first mention) 

4. line 245& 250: strain (not italicized) 

5. line 250: multidrug-resistant (MDR) 

6. line 263: N. gonorrhoeae 

7. line 267: MDR gonococcal lineage 

8. line 279 “bla OXA-580, bla OXA-461, and bla OXA-193”: the gene should be italicized and “OXA-461” should be lower case . This point should be revised through the manuscript. 

9. line 282:MDR strains 

10. line 282: please write the bacterial species in full at the first mention 

11. LINE 285: delete (whole genome sequencing) and write only the abbreviation (WGS) as it was mentioned in the previous sections. 

12. line 286: Klebsiella pneumoniae 

13. line 298-290: and five were pandrug resistant 

14. line 290:an updated reference should be added (Moreover, the first emergence of colistin mcr-10 and fosfomycin fosA5 resistance genes in K.pneumoniae isolated from bovine milk in Egypt was identified using WGS analysis “https://doi.org/10.3389/fmicb.2021.770813”) 

15. line 299: replace “WGC” with “WGS” 

16. line 326: OprD “should be italic.” 

17. line 330-331& 337&341&350” Salmonella typhimurium”: the word “Typhimurium” should start with a capital letter and not be italicized. 2 

18. line 381 “TCA”: write in full at the first mention 

19. line 405: MDR isolates 

20. line 411: S. Typhimurium, K. pneumoniae, S. aureus 

21. line 490& 648: P. aeruginosa 

22. line 499: E. coli 

23. References: 

All bacterial species should be italicized. 

Reference 13& 14: write the pages and volume 

Round 2

Reviewer 2 Report

The author has carefully revised the manuscript according to the suggestions, and I think it can be accepted for publication.